# Social and Emotional Skills Predict Postsecondary Enrollment and Retention

**DOI:** 10.3390/jintelligence11100186

**Published:** 2023-09-22

**Authors:** Kate E. Walton, Jeff Allen, Maxwell J. Box, Dana Murano, Jeremy Burrus

**Affiliations:** 1Behavior and Skills Measurement, ACT, Inc., Iowa City, IA 52243, USA; jeff.allen@act.org (J.A.); dana.murano@act.org (D.M.); jeremy.burrus@act.org (J.B.); 2Department of Psychology, Bowling Green State University, Bowling Green, OH 43403, USA; boxm@bgsu.edu

**Keywords:** social emotional learning, higher education, enrollment, retention

## Abstract

Introduction. Social and emotional (SE) skills are known to be linked to important life outcomes, many of which fall into the academic domain. For example, meta-analytic data show that the skill of Sustaining Effort is nearly or just as important for academic performance as intelligence. In a recent study with long-term tracking of high school students, those who came from schools with a strong emphasis on SE skill development were more likely to enroll in college within two years of high school graduation. Longitudinal studies like this one are rare, however. Method. The focus of the present study is on the SE skills of 6662 students assessed during high school and their relationship with high school academic performance, standardized college admissions test performance, and ultimately postsecondary enrollment and retention. Results. We examined mean-level differences in household income, high school GPA, ACT Composite scores, and SE skills by college enrollment and retention status and found several significant differences, often favoring the enrolled or retained group. Moreover, we found support for the incremental validity of SE skills as they predicted enrollment and retention above household income, high school GPA, and ACT scores. Discussion. Understanding SE skills’ effects on later academic outcomes is important to help inform early SE skill intervention and development efforts in secondary and postsecondary settings. Additional implications and future directions are discussed.

## 1. Introduction

Although academic and demographic factors traditionally have been the focus of research on predictors of academic performance, we now know that social and emotional (SE) skills also play an important role (e.g., [24]; [33]). SE skills have been defined as, “individual capacities that (a) are manifested in consistent patterns of thoughts, feelings, and behaviours, (b) can be developed through formal and informal learning experiences, and (c) influence important socioeconomic outcomes throughout the individual’s life” ([31]). More generally, social and emotional learning (SEL) has been defined as, “the process through which all young people and adults acquire and apply the knowledge, skills, and attitudes to develop healthy identities, manage emotions and achieve personal and collective goals, feel and show empathy for others, establish and maintain supportive relationships, and make responsible and caring decisions ([9]). Furthermore, [9] ([9]) has identified five SE skills that are taught in SEL programs. These include (1) self-awareness—the ability to understand one’s own thoughts and emotions; (2) self-management—the ability to manage one’s emotions, thoughts, and behaviors; (3) social awareness—the ability to understand the perspectives of others and empathize with them; (4) relationship skills—the ability to establish and maintain healthy and supportive relationships; and (5) responsible decision making—the ability to make caring and constructive choices. 

As an example of important outcomes influenced by SE skills, the results of one longitudinal study tracking individuals from Kindergarten through age 25 confirmed that childhood SE skills were negatively associated with public assistance outcomes, such as needing public housing and receiving public assistance, and justice system outcomes, such as having involvement with police before adulthood and being in a detention facility ([19]). Conversely, Kindergarten prosocial skills were significantly and uniquely related to educational and employment outcomes including timely high school graduation, obtaining a college degree, and obtaining stable and full-time employment. 

While there are ample data speaking to the importance of strong SE skills for success in multiple domains of life (e.g., [6]; [23]), longitudinal studies such as that cited above (i.e., [19]) are rare. While the number of studies with longitudinal designs is limited, their results do converge on the importance of SE skills for outcomes throughout the lifespan (e.g., [20]). Understanding SE skills’ effects on later academic outcomes is important to help inform early SE skill intervention and development efforts. The focus of the present study is on SE skills assessed during high school and their relationship with high school academic performance, standardized college admissions test performance, and ultimately postsecondary enrollment and retention. 

## 2. Social and Emotional Skills’ Associations with Key Factors and Outcomes

Prior to engaging in a review of the literature highlighting SE skills’ connections with key academic-related outcomes, we note that the majority of the relevant literature discusses “personality traits” versus “SE skills.” There is a slight yet meaningful distinction between personality traits and SE skills; traits refer to the *tendency* to think, feel, or behave in a particular way while skills refer to the *ability* to think, feel, or behave in a particular way ([39]). Nevertheless, the two constructs have a great deal of overlap with trait–skill pair correlations reaching an average of 0.84 (range = 0.82–0.86; [40]). In fact, a growing number of researchers are arguing that the most robust personality trait framework, the Big Five framework, can be used to organize SE skills ([1]; [10]; [34]; [39]; [47]). The SE skill assessment used in this study, Mosaic^TM^ by ACT^®^ Social Emotional Learning Assessment ([2]), is one of several SE skills assessments utilizing the Big Five framework as an assessment framework. Refer to Table 1 for an illustration of how many common SE skills and the five CASEL competencies ([48]) align with the Big Five traits and the crosswalk between the Big Five and the SE skill terminology used in the Mosaic assessment. Given the great deal of overlap, the forthcoming literature review will cover research in the field of personality psychology as well as SEL, and we will provide both SE skill terms as well as personality trait terms. For example, we will use “Sustaining Effort” as well as “Conscientiousness.”

It is important to note that, while the Big Five framework does not cover every possible SE skill, it does encompass an impressive array of these skills. For example, [47] ([47]) had subject matter experts rate the relatedness of 20 SE skills to each of the Big Five traits. They found that experts rated each of the skills to be at least somewhat related to at least one of the Big Five traits, with nearly every trait judged as moderately to extremely related to at least one trait. Because a study that includes all SE skills identified in the literature would be nearly impossible (at least 136 SE skill frameworks have been identified; Berg et al. 2017), we use the Big Five as our organizing framework as a way to maximize the breadth of SE skills included in our study. 

### 2.1. Academic Performance

Given the abundance of data speaking to the importance of strong SE skills in the academic domain, at least two meta-analyses have been published summarizing the large body of evidence. [33] ([33]) carried out a meta-analysis of Big Five correlations with academic performance. He reviewed 35 studies with nearly 32,000 high school students and reported an average correlation of 0.21 between academic performance and Conscientiousness/Sustaining Effort. As a point of reference, he reviewed 17 studies with more than 12,000 high school students that reported correlations between academic performance and intelligence, and the average correlation reached 0.24. That is, Conscientiousness/Sustaining Effort is nearly or just as important for academic success as intelligence. More recently, [24] ([24]) completed an updated meta-analysis with 267 independent samples and more than 400,000 cases tackling the same research question. He reported that Conscientiousness/Sustaining Effort accounts for 28% of the explained variance in academic performance even when controlling for cognitive ability, demonstrating robust incremental validity of SE skills over cognitive ability in predicting academic performance. In secondary education, corrected correlation coefficients (with academic performance) for Conscientiousness/Sustaining Effort, Openness to Experience/Keeping an Open Mind, and Agreeableness/Getting Along with Others were estimated at 0.27, 0.22, and 0.08, respectively. The estimates for Emotional Stability/Maintaining Composure and Extraversion/Social Connection were near zero. 

The relationship between SE skills and academic performance is likely due to a mediating effect of study habits and skills and attitudes toward academics. For example, those who are higher in Conscientiousness/Sustaining Effort are more likely to complete work on time, check work for mistakes, persist until mastering class material, etc., and therefore earn better marks in school ([11]; see also [49]). 

### 2.2. Standardized College Admissions Tests

[29] ([29]) collected SAT scores from four samples and studied their associations with the Big Five. Openness to Experience/Keeping an Open Mind was consistently correlated with SAT verbal scores while no SE skills were associated with SAT math scores. ACT scores have also been shown to be related to SE skills. In one study, [5] ([5]) reported the greatest effects for Conscientiousness/Sustaining Effort with a correlation between Conscientiousness/Sustaining Effort and ACT Composite scores of 0.27. Students in the highest quartile of Conscientiousness/Sustaining Effort performance scored more than 4 points higher (nearly a full standard deviation) on the ACT than students in the bottom quartile. 

Again, the habits of students high in Conscientiousness/Sustaining Effort (e.g., completing work on time, checking work for mistakes, persisting until mastering class material) likely help students master material to be found on standardized college admissions tests. Likewise, characteristics of students with high Openness to Experience/Keeping an Open Mind scores, such as cognitive flexibility and complexity, lend themselves to stronger performance on these tests.

### 2.3. Postsecondary Enrollment and Retention

In a recent study with long-term tracking of high school students through postsecondary, those who came from schools with a strong emphasis on SE skill development were more likely to enroll in college within two years of high school graduation ([18]). Once enrolled in a postsecondary institution, students with strong SE skills are more likely to persist in their degree programs, according to meta-analytic findings ([36]). Skills related to Conscientiousness/Sustaining Effort showed the strongest associations with postsecondary retention with correlations reaching or exceeding 0.34. 

It is likely that students with strong SE skills are more likely to attend postsecondary institutions because of some of the mechanisms discussed above; these students are more likely to have strong study habits and skills, attitudes towards school, and academic and standardized test performance. In addition, students with strong SE skills have better attendance and fewer disciplinary incidents in high school ([2]; [22]; [43]; [44]). 

### 2.4. Contributions of the Current Study

In addition to examining SE skills’ influence on high school GPA (HSGPA) and ACT Composite scores, our primary objective was to determine the prospective relationship between SE skills measured in high school and postsecondary enrollment and retention. [36]’s ([36]) large meta-analysis included only concurrent data, specifically studies of full-time students already enrolled in a four-year postsecondary institution. Although [18]’s ([18]) study was longitudinal, examining whether and how SE skills measured in high school affect later college enrollment, their data were limited to students from Chicago Public Schools. The generalizability of these data is questionable given that roughly 86% of students are from economically disadvantaged households. There is an established link between SES and educational attainment. Individuals from households with higher SES are more likely to obtain a bachelor’s degree or higher ([27]). Limiting the pool to largely disadvantaged students may impact the conclusions; specifically, effects may be underestimated given the restricted range of postsecondary undertakings.

The current study leverages a longitudinal data set with multiple data sources. The sample comprised a demographically and geographically diverse group of students who completed the Mosaic^TM^ by ACT^®^ Social Emotional Learning Assessment (hereafter referred to as Mosaic), an SE skills assessment based on the Big Five framework of SE skills, in high school. The SE skills assessed were: Sustaining Effort, Getting Along with Others, Maintaining Composure, Keeping an Open Mind, and Social Connection (refer to Table 1 for a description of each skill). These students were then tracked longitudinally to obtain their HSGPAs (self-reported at the time of ACT registration), ACT scores (from ACT), and postsecondary enrollment status for their first and second year (from the National Student Clearinghouse (NSC)).

Our primary research question was: Do SE skills provide incremental validity over HSGPA and ACT Composite scores in predicting year 1 postsecondary enrollment, year 2 postsecondary enrollment, and year 1–year 2 postsecondary retention?

## 3. Method

The study was conducted in accordance with the Declaration of Helsinki. The research is exempt from IRB review according to section 46.104 of the Office for Human Research Protections regulations for the protection of human subjects. The exemption is listed in Section C, Part 2, and is specific to educational tests. Our research meets this requirement given the involvement of educational tests and survey procedures, and the identity of the research participants cannot be readily attained.

### 3.1. Participants

The data used in this study describe a cohort of students who completed an SE skills assessment (described below) and took the ACT test while in high school and were then tracked after high school by the NSC. Data were available for 6662 individuals from 112 different schools across the U.S. Demographic data were collected at the time of ACT registration and/or when the students completed the SE skills assessment. Demographic data including gender, race/ethnicity, grade level, and combined parent/caregiver gross income (hereafter referred to as household income) are presented in Table 2. The sample is fairly typical of the national population, limiting concerns about the generalizability of results. For example, the mean household income is approximately USD 80,000 annually, which is slightly higher than the national median income of USD 70,784, [38] ([38]), and the mean ACT Composite score is 20.5, which is close to the 50th percentile nationally ([3]). 

### 3.2. Instruments

#### 3.2.1. Mosaic^TM^ by ACT^®^ Social Emotional Learning Assessment

While in high school during the 2018–2019, 2019–2020, or 2020–2021 school years, students took the Mosaic^TM^ by ACT^®^ Social Emotional Learning Assessment ([2]). The students were in grades 9 (35%), 10 (24%), 11 (26%), or 12 (15%) when they took the assessment. The students were in schools that were users of Mosaic; that is, students did not self-select to take the assessment. The students took the assessment online in school and typically completed it within a single class period. The assessment measures the five SE skills listed and described in Table 1. Mosaic contains three different item types—Likert items, situational judgment tests, and forced-choice items. The three item types are scored individually (i.e., the mean response per skill per item type is computed), then *z*-scored to account for their being on different metrics, and then the *z*-scores are averaged to arrive at a single score per skill. Readers can refer to the Mosaic technical manual for more details ([2]). Cronbach’s alpha values for the aggregate scores are as follows: Sustaining Effort = 0.70, Getting Along with Others = 0.76, Maintaining Composure = 0.61, Keeping an Open Mind = 0.66, and Social Connection = 0.65. Cronbach’s alpha values for the individual item types that comprise the aggregate scores can be found in the technical manual. 

#### 3.2.2. ACT Scores and GPA

These high school students also took the ACT, and we analyzed their Composite scores (the average of the English, math, reading, and science scores, each on a 1–36 scale). The median score for this sample was 20 (*SD* = 5.68). At the time of ACT registration, they were asked to report their course grades, and these were transformed into a 0–4.0 GPA scale. The median high school GPA (HSGPA) for this sample was 3.57 (*SD* = 0.66). GPA information was missing, however, for 1,860 students.

#### 3.2.3. College Enrollment

We obtained data from the NSC, allowing us to determine whether the students were enrolled in a postsecondary institution one and two years following their high school graduation. This resulted in a binary variable (not enrolled vs. enrolled) for each year[note 1]. Of the 6662 cases, 62.1% were enrolled during their first year out of high school. Considering the 849 students for whom we have year 2 data, 52.1% were enrolled during their second year out of high school. Year 1–year 2 retention data (not retained vs. retained) were available for 531 students, and 77.0% of them were re-enrolled in year 2. 

### 3.3. Data Analysis

#### Data Imputation

There was a substantial amount of data missing for the HSGPA, household income variables, and year 2 enrollment; therefore, we imputed data for these variables to make use of all available data. Nested multiple imputation was conducted using the *mitml* package (v.0.4-5; [16]) in R and a Markov chain Monte Carlo algorithm ([8]; [14]) with high school as a clustering variable (random intercept). A total of 30,000 burn-in iterations, followed by 100 iterations, preceded the imputation of 100 complete datasets using the imputed values. Income, GPA, and year 2 enrollment were imputed using ACT Composite scores, enrollment, gender (male/female), and SE skills as predictors. Race and gender were excluded for convergence purposes. GPA values were constrained to values between 0 and 4, and income was constrained to values between 1 and 9. All continuous variables were subsequently scaled for ease of interpretation in hierarchical mixed effects logistic regression. R-hat values for all chaining parameters fell below 1.05, indicating convergence. Year 2 enrollment appeared to fit well, according to the corresponding FMI values in the subsequent analysis using the benchmark of FMI/m ≤ 0.01 per predictor.

Before imputation, the mean HSGPA was 3.37 (*SD* = 0.66) and after imputation it was 3.31 (*SD* = 0.65). Before imputation, the mean household income was 5.34 (*SD* = 2.68) and after imputation it was 5.28 (*SD* = 2.57). Values of 5 and 6 correspond to “About USD 60,000-80,000” and “About USD 80,000-100,000”, respectively.

We carried out a series of independent sample *t*-tests to determine whether there were significant (*p* < .05) mean-level differences between students enrolled in year 1 and those not enrolled, between students enrolled in year 2 and those not enrolled, and students who retained enrollment from year 1 to year 2 and those who did not. We examined differences between household income, GPA, ACT Composite, and SE skill scores. Standardized effect sizes (Cohen’s *d*) were calculated for each test.

To answer our primary research question, we fit a series of hierarchical mixed effects logistic regression models, regressing household income (step 1)[note 2], HSGPA and ACT Composite scores (step 2), and the five SE skills (step 3) on year 1 enrollment, year 2 enrollment, and year 1–year 2 retention. School membership was used as a clustering variable using the pooled datasets via “Rubin’s Rules” ([37]). Models constructed from the pooled datasets were then compared using the method provided by [21] ([21]). Accuracy of classification was determined by averaging the proportion of cases correctly classified after prediction using pooled parameter estimates and a fixed intercept.

## 4. Analyses and Results

### 4.1. Descriptive Statistics and Group Mean Differences

Correlations among all continuous variables can be found in Table 3. Results pertaining to mean-level comparisons of those enrolled or retained vs. not can be found in Table 4. The amount of power to detect effect sizes of 0.20, 0.50, and 0.80, respectively, for year 1 enrollment is 1.00, 1,00, and 1.00. For year 2 enrollment, these values are 0.83, 1.00, and 1.00, and for retention, they are 0.49, 1.00, and 1.00. 

Year 1 enrolled students, year 2 enrolled students, and year 1–year 2 retained students had statistically significantly higher household incomes than those not enrolled or retained. Effect sizes reached 0.46 for the year 2 enrolled vs. not enrolled test.

Year 1 enrolled students, year 2 enrolled students, and year 1–year 2 retained students had statistically significantly higher GPAs and ACT Composite scores than their counterparts. Effect sizes reached 1.00 for GPA and 0.87 for ACT scores for the year 2 enrolled vs. not enrolled test. The effects were slightly lower for year 1 enrolled vs. not enrolled and retained vs. not retained tests. 

For SE skills, the largest effects were observed for Sustaining Effort. Year 1 enrolled students, year 2 enrolled students, and year 1–year 2 retained students had statistically significantly higher Sustaining Effort scores than their counterparts. Year 1 enrolled students had statistically significantly higher Getting Along with Others, Maintaining Composure, Keeping an Open Mind, and Social Connection scores than their counterparts. Year 2 enrolled students had statistically significantly higher Getting Along with Others scores than individuals not enrolled in year 2. The direction of mean differences for retention was different for two SE skills, namely, Maintaining Composure (*d* = −0.22) and Keeping an Open Mind (*d* = −0.27). That is, students who did not maintain enrollment status from year 1 to year 2 were higher on these two skills.

### 4.2. Hierarchical Logistic Regression Models

We next fit the regression models to evaluate the incremental validity of SE skills. With each outcome variable, the change in χ^2^ at each step was statistically significant (Table 5), indicating the addition of the variables significantly improved the fit of the model. The final model statistics for each outcome can be found in Table 6. Note that we report standardized OR to enable us to compare the relative importance of the predictor variables. We opted to interpret results in terms of the magnitude of the effect rather than simply statistical significance. Looking across the three outcomes, note that the sample size drove statistical significance (e.g., an OR of 1.30 would be significant for year 1 but not for year 2 enrollment); therefore, albeit somewhat arbitrary, here we highlight any OR ≤ 0.90 or ≥1.10.

Higher household income, HSGPA, and ACT Composite, Sustaining Effort, and Getting Along with Others scores were associated with greater odds of year 1 enrollment. HSGPA was the biggest determinant of year 1 enrollment; with a one-standard deviation increase in HSGPA, the odds of enrolling in college increased 71%. Importantly, one SE skill—namely, Sustaining Effort (OR = 1.27)—had a stronger effect than household income (OR = 1.16). Nearly three quarters (71.2%) of the sample was correctly identified as being enrolled or not based on estimated probabilities. For each instance, if the estimated probability exceeded 0.50, the case was classified as enrolled. Recall that 62.1% of the sample was enrolled in year 1.

All variables except for Social Connection had sizable effects on year 2 enrollment. Higher household income, HSPGA, ACT, Sustaining Effort, and Getting Along with Others scores were associated with increased odds of year 2 enrollment. Higher Maintaining Composure and Keeping an Open Mind were associated with reduced odds of year 2 enrollment. HSGPA was the biggest determinant of year 2 enrollment; with a one-standard deviation increase in HSGPA, the odds of enrolling in college increased 147%. Two SE skills had stronger effects than household income. Nearly three quarters (70.3%) of the sample was correctly identified as being enrolled or not based on estimated probabilities. Recall that 52.1% of the sample was enrolled in year 2.

All variables except for Social Connection had sizable effects for year 1–year 2 retention. Higher household income, HSPGA, ACT, Sustaining Effort, and Getting Along with Others scores were associated with increased odds of retention. Higher Maintaining Composure and Keeping an Open Mind were associated with reduced odds of year 2 enrollment. Again, two SE skills had stronger effects than household income. More than 68% of the sample was correctly identified as retaining enrollment status year 1–year 2 (68.3%). This was a decrease from the 77.0% who were retained, though four variables’ estimated coefficients were statistically significant.

## 5. Discussion

SEL is growing at an increasingly fast rate, and the necessity for strong SE skills among students, employees, and citizens in general is widely acknowledged. For example, a recent survey of companies reveals that Big Five-related skills are among the top ten required skills of workers, topping reading, writing, and mathematics skills on the list ([50]). There is a growing body of literature showing the importance of SE skills for important outcomes, including academic performance ([24]). Less is known about the long-term effects of SE skills, including their effects on educational attainment; thus, our goal with the current study was to examine the effects of high school students’ SE skills on schoolwork, a standardized college admissions test performance, and ultimately, postsecondary enrollment and retention. This is critical because postsecondary education is associated with a host of positive outcomes, ranging from greater chances of employment ([15]) and higher earnings ([17]) to better health ([13]). What is more, the effects extend beyond the individual level. According to the [30] ([30]), individuals with higher education can be expected to earn, on average, 55% more than individuals with no higher education. The societal-level return on investment is estimated at $91,000 per student for OECD countries. Therefore, any and all factors—SE skills included—that positively influence educational attainment should be identified and then fostered. 

### 5.1. Overview of Current Study and Findings

In this study, we leveraged a rich dataset of 6662 students from 112 diverse schools across the nation. We built a holistic and long-range view of the students, compiling SE skills data obtained in grades 9–12, ACT-provided SES information, HSGPA, and ACT Composite scores, and NSC-provided information on postsecondary enrollment one and two years past high school. Our goal was to use this rich dataset to examine the role SE skills play in postsecondary enrollment and retention when considered among other factors typically associated with postsecondary enrollment—standardized test scores, GPA, and SES. 

First, we examined associations between SE skills and academic and college admissions test performance. Consistent with prior meta-analytic research ([24]; [33]), Sustaining Effort was the SE skill with the strongest association with HSGPA. The correlation reached 0.43 in our sample, 0.27 in Mammadov’s meta-analysis, and 0.21 in Poropat’s meta-analysis. Overall, our SE skill-HSGPA correlations were higher than reported previously. This may be because the research included in the meta-analyses was specific to “personality traits”, as opposed to “SE skills”. The items on the Mosaic assessment are likely more contextualized to the school setting than a typical Big Five personality measure. 

With regard to college admissions tests, specifically the SAT, [29] ([29]) reported near-zero correlations with SAT math across all Big Five personality traits but concluded that Openness is significantly related to SAT verbal scores with correlations reaching 0.26 in two of their four samples. In contrast, we found that Sustaining Effort had the strongest relationship with ACT scores (*r* = 0.27), followed by Keeping an Open Mind (*r* = 0.18), and Getting Along with Others (*r* = 0.17). Two factors may account for this difference. First, as mentioned above, the Mosaic items are more contextualized to an academic setting as opposed to typical Big Five personality measures such as the ones used in Noftle and Robins’s study. Second, we analyzed ACT Composite scores, which, include math, science, English, and reading, compared with SAT scores that only contain math and verbal components. It is possible that science success is more reliant on Sustaining Effort than on Keeping an Open Mind.

We found support for our primary research question pertaining to whether SE skills would provide incremental validity in predicting postsecondary enrollment and retention above household income, HSGPA, and ACT scores, which are known quantities in terms of predicting postsecondary enrollment (e.g., [32]). In three series of hierarchical regressions, model fit improved upon the entry of SE skills into the models after household income, HSGPA, and ACT Composite scores, indicating clearly that SE skills are important predictors of college enrollment and retention. This result is in line with past findings indicating the importance of non-cognitive predictors of retention in postsecondary settings, such as emotional intelligence (e.g., [41]). 

We examined specifically which skills played the most prominent role in predicting the outcomes. Across the three dependent variables, Getting Along with Others was a significant positive predictor, consistent with prior research ([35]). That is, being empathic, trusting, trustworthy, and a good collaborator are all linked with increased odds of enrolling in a postsecondary institution one and two years after high school and maintaining a positive enrollment status from year 1 to year 2. High Sustaining Effort was the strongest SE skill in predicting year 1 and year 2 enrollment but did not play a significant role in retention. Social Connection was a predictor for retention only, with higher scores being associated with greater odds of retention. It stands to reason that greater social connection and vitality would contribute to one’s desire to remain in the current postsecondary institution. Prior meta-analytic data have also shown that SE skills are related to postsecondary retention ([36]). Those with the largest effects are academic-related skills (e.g., time management, leadership skills, problem solving, coping strategies), academic self-efficacy, and academic goals (e.g., persistence and goal-directed behavior). Furthermore, these effects were on par with or greater than the effects for HSGPA and SES, which is similar to our findings reported here. 

Initially, we were surprised by the Sustaining Effort finding in that the skill only predicted enrollment, but not retention from year 1 to year 2. Among further consideration though, it seems reasonable that the initial process of successfully applying to and beginning college is more dependent on skills associated with Sustaining Effort, such as being goal-oriented, remaining organized and managing deadlines, and completing processes. The decision to remain in a university from year 1 to year 2, however, may be more contingent on interpersonal factors and student–institution fit ([26]). 

Interestingly, Maintaining Composure and Keeping an Open Mind was associated with year 2 enrollment and retention but in the opposite direction; that is, higher scores for Maintaining Composure and Keeping an Open Mind were associated with lower odds of year 2 enrollment and retention. There is some evidence that Maintaining Composure has a curvilinear relationship with postsecondary outcomes. Specifically, [35] ([35]) found that it had a curvilinear U relationship with GPA. We can surmise that a similar pattern might occur for retention where hypo- and hyper-control could be detrimental to one’s commitment to postsecondary education. For example, perhaps students with extremely high levels of Maintaining Composure become overwhelmed with the stress associated with the schoolwork and feel the need to withdraw, either permanently or temporarily, from their institution. Alternatively, perhaps individuals with extremely high levels of Maintaining Composure simply feel comfortable recognizing that their initial decision to attend a postsecondary institution was the wrong decision and withdraw, again either permanently or temporarily, making an unexpected and non-traditional choice. In terms of Keeping an Open Mind, we can also conjecture as to why higher scores are associated with reduced odds of year 2 enrollment and year 1–year 2 retention. There are two primary facets of Keeping an Open Mind, one that pertains more to intellect (an example item is: *I am quick to understand things*) and one that pertains more to openness (an example item is: *Believe in the importance of art*; [12]). It stands to reason that intellect would be a positive predictor of enrollment and retention, but the direction of the effect of openness is less clear. Of note, the items in Mosaic are more strongly reflective of the latter, more general openness facet. Perhaps individuals high on openness are more open to trying a different path and, therefore, are more likely to withdraw from their postsecondary institutions. 

### 5.2. Implications

As expected, students from wealthier families and who have higher HSGPAs and ACT Composite scores were more likely to be enrolled in year 1 and year 2 and to retain their status of being enrolled in a postsecondary institution from year 1 to year 2. We find it remarkable that in each instance of the three dependent variables, one or more SE skills had larger effects than that for household income. Clearly, household income is a fixed value that cannot be manipulated by educators. SE skills, in contrast, can be developed. There are at least four meta-analyses summarizing the effects of SEL programming, all of which show positive effects in multiple domains ([23]). Students involved in SEL showed lower levels of emotional distress and conduct problems and stronger SE skills and academic performance. What is more, the academic improvements translate into an 11-percentile-point gain in achievement. Importantly, Mahoney and colleagues argue that SEL programming clearly fosters academic success, rather than detracting from it. Our findings echo this sentiment; many SE skills are associated with being on track for tertiary degree completion. The malleability of SE skills, given their relevance in enrollment and persistence, is particularly important for students from lower-income households and historically underrepresented groups, given persistence rates can be as low as 50% in terms of students falling into one or more of these categories, while they are closer to 80% for students from higher income, White, and Asian families ([28]).

In terms of the implications of our findings, there is a second important point to consider. ACT Composite scores were a stronger predictor of enrollment and retention. For example, the standardized effect size comparing ACT scores of those enrolled in year 2 and those not enrolled was 0.87. As is well known, some postsecondary institutions are instituting test optional or test-blind policies. One of the most well-known instances of this practice is carried out by the University of California system whose policy is to “… not consider SAT or ACT test scores when making admissions decisions or awarding scholarships” ([46]). If we lose those data that can be used to predict enrollment and retention, we need other means of anticipating students’ behaviors and trajectories. We have shown here that, coupled with high school academic performance, SE skills data can provide us with valuable data in predicting enrollment and retention. While the current study shows that measuring SE skills improves prediction and the identification of at-risk students, the greatest benefit occurs when the measurement of SE skills leads to improvements in the effectiveness of interventions ([4]).

### 5.3. Limitations and Future Directions 

One limitation of this work may be the use of the Big Five framework as the organizing framework for SE skills. Although we and others have argued that the Big Five is a strong, and possibly the best, framework for organizing SE skills (e.g., [25]; [47]), there is a multitude of frameworks that could be studied ([7]). It is possible that different results could be found for the SE skills found in these alternative frameworks. We chose to use the Big Five in our research because it is a parsimonious framework that encompasses a broad set of SE skills, and it is simply not possible to conduct a research study utilizing every possible SEL framework. However, future research can attempt to replicate our findings using the constructs included in different frameworks as predictor variables. 

Furthermore, our data set was limited in terms of its long-term reach. We plan to continue to monitor these students and seek data from the NSC to follow these students further into their postsecondary careers. We also have SE skills assessments for elementary and middle schools, so the long-term view of students can only grow. Second, we hope to collect data confirming which students did and did not participate in SEL programming in their schools to determine if that has effects, whether direct or indirect, on postsecondary enrollment and retention. Third, our models only considered the incremental direct effect of SE skills on college enrollment and retention. Given that SE skills also influence HSGPA and ACT scores (e.g., [42]), the total effects of SE skills on college enrollment and retention are likely much larger. Finally, our findings show that Maintaining Composure and Keeping an Open Mind had negative effects on the outcome variables. We were able to merely conjecture as to why this may be, but future research should serve to further investigate this issue.

## 6. Conclusions

Our findings clearly show that early SE skills predict later postsecondary enrollment and retention. Higher education is linked with many positive outcomes; therefore, it is critical to consider all factors that influence students’ educational paths, particularly those that are malleable. SE skills, by definition, “can be developed through formal and informal learning experiences” ([31]). SEL programming has been shown to effectively develop SE skills and has many positive short- and long-term outcomes ([23]). Assessing SE skills during high school (and even earlier) helps predict college readiness and postsecondary pathways, and subsequent curricula can enable students to develop skills needed to achieve in high school, reach college, and then achieve in college and beyond.

## Figures and Tables

**Table 1 jintelligence-11-00186-t001:** Crosswalk between Big Five personality traits and social and emotional skills.

Big Five Factor	Big Five Factor Definition	Common Social and Emotional Skills	CASEL Competencies	Mosaic Social and Emotional Skill Label
Conscientiousness	Describes socially prescribed impulse control that facilitates task- and goal-directed behavior, such as thinking before acting, delaying gratification, following norms and rules, and planning, organizing, and prioritizing tasks.	Attention to Detail; Grit; Impulse Control; and Organization	Self-Management; and Responsible Decision Making	Sustaining Effort
Agreeableness	Implies a prosocial and communal orientation toward others and includes traits such as altruism, tender-mindedness, trust, and modesty.	Collaboration; Empathy; Relationship Skills; and Teamwork	Social Awareness; Relationship Skills; and Responsible Decision Making	Getting Along with Others
Emotional Stability	Implies being emotionally stable and even-tempered, rather than experiencing negative emotionality, such as feeling anxious, nervous, sad, and tense.	Resilience; Self-efficacy; Self-regulation; and Stress management	Self-Awareness; and Self-Management	Maintaining Composure
Openness to Experience	Describes the breadth, depth, originality, and complexity of an individual’s mental and experiential life.	Appreciation for Diversity; Creativity; Curiosity; and Problem solving	Social Awareness	Keeping an Open Mind
Extraversion	Implies an energetic approach towards the social and material world and includes traits such as sociability, activity, assertiveness, and positive emotionality.	Assertiveness; Leadership; Optimism; and Social Engagement	Self-Awareness; and Relationship Skills	Social Connection

**Table 2 jintelligence-11-00186-t002:** Participant demographic data.

	*N*	Valid %
Gender		
Female	3525	53.0
Male	3124	46.9
Other	6	.0
Decline to Respond	7	
Race/Ethnicity		
American Indian/Alaska Native	228	3.6
Asian	211	3.3
Black/African American	416	6.6
Hispanic/Latino	805	12.7
Native Hawaiian/Other Pacific Islander	14	.2
White	4420	69.7
Two or More	247	3.9
Other or Decline to Respond	321	
ACT Test Grade Level		
9th	2	.0
10th	42	1.0
11th	4236	63.8
12th	2356	35.5
Missing or Invalid Response	26	
Household Income		
<USD 24,000	393	10.1
USD 24,000–36,000	393	10.1
USD 36,000–50,000	394	10.1
USD 50,000–60,000	305	7.9
USD 60,000–80,000	453	11.7
USD 80,000–100,000	473	12.2
USD 100,000–120,000	412	10.6
USD 120,000–150,000	352	9.1
>USD 150,000	707	18.2
Missing	2780	

**Table 3 jintelligence-11-00186-t003:** Intervariable correlations.

	Income	HSGPA	ACT	SE	GA	MC	KO	SC
Household Income (Income)								
High School GPA (HSGPA)	0.33							
ACT Composite (ACT)	0.38	0.60						
Sustaining Effort (SE)	0.15	0.43	0.27					
Getting Along with Others (GA)	0.07	0.25	0.17	0.61				
Maintaining Composure (MC)	0.06	0.18	0.13	0.48	0.61			
Keeping an Open Mind (KO)	0.03	0.20	0.18	0.53	0.68	0.57		
Social Connection (SC)	0.04	0.14	0.10	0.47	0.57	0.56	0.63	

*Note.* All correlations were statistically significant at *p* < .01.

**Table 4 jintelligence-11-00186-t004:** Descriptive statistics and group-level mean comparisons.

	Enrolled or Retained	Not Enrolled or Retained			
	*M* (*SD*)	*M* (*SD*)	*t*	*p*	*d*
Household Income					
Year 1	5.60 (2.39)	4.53 (2.34)	17.87	<0.01	0.45
Year 2	5.21 (2.37)	4.15 (2.25)	6.71	<0.01	0.46
Retained	5.29 (2.39)	4.40 (2.24)	3.64	<0.01	0.38
GPA					
Year 1	3.47 (.52)	2.98 (.72)	31.84	<0.01	0.80
Year 2	3.45 (.53)	2.84 (.70)	14.49	<0.01	1.00
Retained	3.47 (.51)	3.09 (.55)	7.05	<0.01	0.73
ACT Composite					
Year 1	21.97 (5.46)	18.11 (5.19)	28.51	<0.01	0.72
Year 2	22.25 (5.47)	17.82 (4.66)	12.67	<0.01	0.87
Retained	22.49 (5.43)	18.96 (4.64)	6.52	<0.01	0.67
Sustaining Effort					
Year 1	0.34 (1.39)	−0.33 (1.32)	19.43	<0.01	0.49
Year 2	0.39 (1.53)	−0.31 (1.39)	6.89	<0.01	0.47
Retained	0.43 (1.53)	0.17 (1.44)	1.72	0.04	0.18
Getting Along with Others					
Year 1	0.38 (1.30)	−0.06 (1.37)	13.03	<0.01	0.33
Year 2	0.50 (1.35)	0.16 (1.73)	3.17	<0.01	0.22
Retained	0.50 (1.34)	0.45 (1.44)	0.38	0.35	0.04
Maintaining Composure					
Year 1	0.13 (1.28)	−0.09 (1.21)	7.09	<0.01	0.18
Year 2	0.16 (1.29)	0.16 (1.64)	0.00	0.99	0.00
Retained	0.16 (1.29)	0.50 (2.23)	−2.13	0.02	−0.22
Keeping an Open Mind					
Year 1	0.11 (1.26)	−0.16 (1.31)	8.31	<0.01	0.21
Year 2	0.03 (1.30)	0.13 (1.37)	−1.06	0.29	−0.07
Retained	0.03 (1.28)	0.37 (1.20)	−2.65	<0.01	−0.27
Social Connection					
Year 1	−0.10 (1.26)	−0.27 (1.24)	5.27	<0.01	0.13
Year 2	−0.01 (1.29)	−0.04 (1.37)	0.31	0.76	0.02
Retained	0.02 (1.24)	0.06 (1.40)	−0.31	0.38	−0.03

**Table 5 jintelligence-11-00186-t005:** Change in model fit statistics.

	Δχ^2^	df	*p*
Year 1 Enrollment			
Predictor Variable(s)			
Caregiver income	--		
HSGPA and ACT Composite	263.46	2	<0.01
SE skills	10.32	5	<0.01

Year 2 Enrollment			
Predictor Variable(s)			
Caregiver income	--		
HSGPA and ACT Composite	77.18	2	<0.01
SE skills	2.69	5	0.02

Retention			
Predictor Variable(s)			
Caregiver income	--		
HSGPA and ACT Composite	131.14	2	<.01
SE skills	2.90	5	.02

**Table 6 jintelligence-11-00186-t006:** Model estimates.

	*B*	*t*	*p*	Standardized OR
Time 1 Enrollment				
Predictor Variable				
Caregiver income	0.15	3.34	<0.01	1.16
HSGPA	0.54	10.71	<0.01	1.71
ACT Composite	0.27	6.01	<0.01	1.32
Sustaining Effort	0.24	5.18	<0.01	1.27
Getting Along with Others	0.12	2.54	0.01	1.13
Maintaining Composure	−0.03	−0.75	0.46	0.97
Keeping an Open Mind	−0.05	−1.04	0.30	0.95
Social Connection	−0.07	−1.71	0.09	0.93

Time 2 Enrollment				
Predictor Variable				
Caregiver income	0.15	1.31	0.19	1.16
HSGPA	0.91	7.12	<0.01	2.47
ACT Composite	0.18	1.68	0.10	1.20
Sustaining Effort	0.10	0.90	0.37	1.10
Getting Along with Others	0.19	2.01	0.05	1.21
Maintaining Composure	−0.13	−1.47	0.14	0.87
Keeping an Open Mind	−0.29	−2.97	<0.01	0.75
Social Connection	0.04	0.50	0.62	1.04

Retention				
Predictor Variable				
Caregiver income	0.15	1.87	0.06	1.16
HSGPA	0.80	8.74	<0.01	2.23
ACT Composite	0.20	2.65	<0.01	1.22
Sustaining Effort	0.13	1.62	0.11	1.13
Getting Along with Others	0.18	2.42	0.02	1.19
Maintaining Composure	−0.12	−1.71	0.09	0.89
Keeping an Open Mind	−0.20	−2.75	<0.01	0.82
Social Connection	0.02	0.30	0.77	1.02

*Note*. For all tests, df = 1.

## Data Availability

The data are proprietary and are therefore not available.

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
