# Peer review of "Social and Emotional Skills Predict Postsecondary Enrollment and Retention"

_jintelligence, 2023, doi:10.3390/jintelligence11100186_

Round 1

Reviewer 1 Report

This manuscript presents a large-scale, longitudinal investigation of the effects of SE skills on postsecondary enrollment and detention. Overall, I believe it will make a valuable addition to the literature. Please see below for some comments that could potentially enhance the manuscript:

1. The authors equate SE skills with personality throughout this manuscript, citing the high correlations reported by Soto and colleagues as supporting evidence. However, in education, many other theoretical SE frameworks and associated measures, which were not built upon the Big Five, exist apart from the BESSI framework proposed by Soto and colleagues. These measures have shown substantially lower correlations with personality. Therefore, equating personality with SE skills may not be the most appropriate approach. While these concepts are related, treating them as equivalent is a bold claim. I recommend the authors reframe the introduction to differentiate between personality and SE skills.

2. The authors used multiple imputation to handle missing data. However, it remains unclear how the authors managed the nested data structure (students nested within schools) during the multiple imputation. If the authors did not consider the nestedness, I recommend they do so. If they have already considered it, I suggest clarifying this in the manuscript.

3. The authors only imputed data from HSGPA and household income. However, the variable "second year enrollment" also had a significant amount of missing data. Why not also impute for this key outcome variable?

4. Picking results at random from a multiply imputed dataset is not recommended as it ignores the uncertainty inherent in missing data. I strongly suggest the authors report results pooled across the five imputed datasets following Meng and Rubin's (1992) pooling procedure.

Enders, C. K., Mistler, S. A., & Keller, B. T. (2016). Multilevel multiple imputation: A review and evaluation of joint modeling and chained equations imputation. Psychological Methods, 21(2), 222.

Meng, X. L., & Rubin, D. B. (1992). Performing likelihood ratio tests with multiply-imputed data sets. Biometrika, 79(1), 103-111.

Sinharay, S., Stern, H. S., & Russell, D. (2001). The use of multiple imputation for the analysis of missing data. Psychological Methods, 6(4), 317–329.

5. When performing the focal analyses, how was the nested data structure handled?

The writing looks ok to me. 

Reviewer 2 Report

The manuscript is sound and clear . I have some comments that might be addressed:

1) The focus of the paper is vague. It seems that the authors attempt to investigate the SE skills.  I suggest to add more information on the type of skills, or the educational settings or the educational approaches to best refer to the findings by the authors.

2) the second, third, and fourth categories in Table 1, when translated to mosaic social and emotional skill labels, seems to not group the common social and emotional skills. I suggest rename it. On the contrary 'sustaining effort' and 'social connection', are clear, concise and technically sound.

3) On Table 3, I suggest the use of the standard asteriscs for the significant correlation, or double asterics in p<0.01.

4) Could you specify what does it mean year1 and year 2 enrollment. For an european researcher this is confusing. Could we know what type of education is received in both years. Is for example year 1 or year 2, more oriented towards some educational approach that explains the results?

Reviewer 3 Report

To be admitted, the article requires changes, which the authors must solve.

1. The abstract must be written in IMRD format (Introduction, Method, Results, Discussion).

2. Remove references and citations from the abstract.

3. Include more details of the most relevant results and the discussion.

4. The keywords do not respond to recognized descriptors, check the keywords used, you can check the ERIC keywords, for example.

5. It is rare to start the article with the key lines before the introduction section.

6. It is essential to report the reliability of the instruments used, at least report the alpha and omega coefficients.

7. The Sections of the Method must be:

- Participants

- Instruments

- Procedure

- Data analysis

8. Adapt the preparation of the article to these sections indicated in the method.

9. There is no information on ethical standards, ethics committees, the Declaration of Helsinki, etc.

10. In the results, it is necessary to report the statistical power (not calculated) and reflect on the high size of the effect in some pairs of comparisons of variables with Cohen's d values >.8

11. In the discussion section, further discussion of the evidence found and its contrast with the results of other studies is necessary. More contrast, more references, more studies, more discussion.

12. 50% of the references are from recent years.

13. But it is necessary to increase the number of citations and references to discuss and substantiate the study in more detail.

Round 2

Reviewer 1 Report

The authors have adequately addressed my previous comments. I only have two minor comments that can be easily addressed. 

Page 4. “In secondary education, corrected correlation coefficients (with academic performance) for Conscientiousness/Sustaining Effort, Openness to Experience/Keeping an Open Mind, and Agreeableness/Getting Along with Others were estimated at .56, .45, and .16, respectively.” These numbers were not correlations. They were Cohen’s d. The corresponding correlations were .27, .22, and .08. Please double-check all numbers cited in the manuscript to make sure they are correctly reported. 

Page 8. I think it is more appropriate to say “in order to make use of all available data” instead of  “increase sample size” when describing the purpose of multiple imputation. 

There are still some typos/grammatical errors. A thorough proofreading should be conducted. 

Author Response

Thank you for the thorough review. These edits have been made.

Reviewer 3 Report

The authors have made all the suggested changes and the manuscript has been greatly improved.

I consider that the article meets the requirements to be published.

All the changes are detailed and all the modifications made can be checked in detail.

Author Response

Thank you for the thorough review.

Round 3

Reviewer 1 Report

The authors have adequately addressed my comments.